# Molecular Insights into Ammonium Sulfate-Induced Secretome Reprogramming of *Bacillus subtilis* Czk1 for Enhanced Biocontrol Against Rubber Tree Root Rot

**DOI:** 10.3390/microorganisms13092212

**Published:** 2025-09-21

**Authors:** Yanqiong Liang, Shibei Tan, Ying Lu, Helong Chen, Xing Huang, Kexian Yi, Chunping He, Weihuai Wu

**Affiliations:** 1Environment and Plant Protection Institute, Chinese Academy of Tropical Agricultural Sciences, Haikou 571101, China; yanqiongliang@126.com (Y.L.); tanshibei915@163.com (S.T.); ytluy2010@163.com (Y.L.); chenhelong951@126.com (H.C.); hxalong@gmail.com (X.H.); 2Sanya Research Institute of Chinese Academy of Tropical Agricultural Sciences, Sanya 572025, China; yikexian@126.com; 3Key Laboratory of Integrated Pest Management on Tropical Crops, Ministry of Agriculture and Rural Affairs, Haikou 571101, China; 4Hainan Key Laboratory for Detection and Control of Tropical Agricultural Pests, Haikou 571101, China; 5Hainan Engineering Research Center for Biological Control of Tropical Crops Diseases and Insect Pests, Haikou 571101, China

**Keywords:** rubber tree root rot, label-free quantitative proteomics, *Bacillus subtilis* Czk1, proteomics

## Abstract

Root rot diseases caused by *Ganoderma pseudoferreum* and *Pyrrhoderma noxium* inflict substantial economic losses in rubber tree (*Hevea brasiliensis*) cultivation, while conventional control methods face environmental and resistance challenges. This study aimed to specifically investigate the molecular mechanisms by which ammonium sulfate enhances the biocontrol efficacy of *Bacillus subtilis* Czk1. Using label-free quantitative proteomics (LC-MS/MS), we characterized ammonium sulfate-induced alterations in the secretory proteome of Czk1. A total of 351 differentially expressed proteins (DEPs) were identified, with 329 significantly up-regulated and 22 down-regulated. GO functional enrichment analysis indicated that up-regulated DEPs were associated with metabolic pathways (glyoxylate/dicarboxylate, arginine/proline, cofactor biosynthesis) and extracellular localization (13 proteins), while down-regulated DEPs were linked to small molecule catabolism. KEGG pathway annotation identified DEP involvement in 124 pathways, including secondary metabolite biosynthesis and membrane transport. These findings demonstrate that ammonium sulfate remodels the Czk1 secretome to enhance the expression of key antagonistic proteins, thereby providing crucial molecular targets and a scientific foundation for developing effective biofungicides against rubber root rot, with clear practical implications for sustainable disease management.

## 1. Introduction

The rubber tree (*Hevea brasiliensis* Muell. Arg.), a perennial tropical crop, is a key source of high-quality natural rubber. It serves as a vital economic pillar for natural rubber production and significantly contributes to the livelihoods of smallholder farmers in developing countries [1,2,3]. Root rot disease is one of the three most devastating diseases in natural rubber production and is serious and widespread in rubber-growing areas around the world. It is also a major biological constraint limiting rubber yield growth in China [4]. A total of seven distinct types of rubber root diseases have been documented in China, including red root rot disease (*Ganoderma pseudoferreum* (Wakef.) Over et Steinm.), brown root rot disease (*Pyrrhoderma noxium* (Corner) L.W. Zhou & Y.C. Dai), purple root rot disease (*Helicobasidium* compactum Boed), stink root rot disease (*Sphaerostilbe repens* Berk. et Br.), white root rot disease (*Rigidoprus lignosus* (Klotzsch) Imaz), black root rot disease (*Poria hypobrunnea* Petch.) and ustulina root rot disease (*Ustulina deusta* (Hoffmm et Fr.) Petrak), of which red root rot and brown root rot are the most harmful [5]. Among these, red root rot and brown root rot are prevalent in various rubber plantation areas in China. These two diseases are prevalent across Chinese rubber plantations, with severe infection prevalence exceeding 4.0%. Without timely intervention, mortality rates can reach 100% [6,7,8,9].

The diagnosis of rubber tree root disease remains heavily labor-dependent. Early detection systems are often inadequate, resulting in inefficient monitoring and delayed responses. By the time symptoms become visible in the aboveground parts of the tree, the infection is usually already severe, leading to high control costs and limited effectiveness. At present, the primary management strategies include agricultural and chemical controls. Agricultural methods mainly consist of trench isolation, which serves only as a preventive measure and is both time-consuming and labor-intensive. Chemical control has long relied on the application of thirteen morpholine for root irrigation [10,11,12,13]. However, this approach poses significant risks such as pathogen resistance, pesticide residues, and environmental pollution.

Biological control is an eco-friendly strategy that employs naturally occurring microorganisms and their metabolites to manage plant diseases [14,15]. The metabolites produced by biocontrol bacteria are relatively stable in the environment and less influenced by external conditions, making them among the most successfully commercialized biocontrol products to date. The use of antimicrobial compounds derived from microorganisms thus represents a pioneering green approach to plant disease management. Numerous microorganisms have been exploited for the development of biopesticides on a commercial scale. Among these, *Bacillus* spp. have proven to be a highly effective source of antifungal agents and are widely employed as biocontrol organisms [16,17,18]. A key advantage of *Bacillus* spp. lies in their ability to produce diverse bioactive compounds with agricultural applications, including antimicrobial metabolites, surfactants, and plant defense stimulators [19,20,21].

Proteomics, an integral component of functional genomics, enables the isolation of protein complexes and functional characterization, offering critical insights into biological systems [22,23]. Among its advanced methodologies, label-free quantitative proteomics stands out as a cost-effective and scalable approach. This technique leverages modern high-resolution mass spectrometry to analyze enzymatically digested peptide mixtures without isotope labeling, thereby facilitating large-scale comparative studies of proteome dynamics [24,25]. This technique has been widely applied to dissect molecular mechanisms across disciplines. In plant–pathogen interactions, for instance, Yu et al. (2019) demonstrated that nitrogen starvation in Phytophthora infestans—the pathogen responsible for potato late blight—upregulated virulence-related effector proteins, thereby enhancing pathogenicity [26]. In abiotic stress studies, Liu et al. (2021) identified N-glycans as critical modulators of salt stress tolerance in Arabidopsis through regulation of stress-responsive proteins [27]. Similarly, Song et al. (2022) revealed proteomic adaptations in Dendrobium huoshanense under salt stress, highlighting key proteins involved in osmotic adjustment and antioxidant defense [28]. Beyond plant biology, label-free proteomics has advanced applications in food science: Zhang et al. (2024) systematically linked freezing–thawing methods (e.g., liquid nitrogen, immersion, and air freezing) to protein denaturation and quality deterioration in perch filets [29], whereas Menezes et al. (2024) demonstrated that sourdough fermentation significantly reduces allergenic wheat components, such as glutenins and α-amylase inhibitors [30]. Collectively, these applications highlight the versatility and impact of label-free proteomics in advancing molecular understanding and supporting innovation across biological and industrial fields.

The *Bacillus subtilis* Czk1 strain, isolated from the roots of the rubber tree in our laboratory, exhibits potent antifungal activity against several major pathogens, including *G. pseudoferreum*, *P. noxium*, *H. compactum*, *R. lignosus*, *S. repens* and *C. gloeosporioides* [31,32]. To characterize the protein composition of its fermentation broth and investigate the molecular basis of its biocontrol activity, we employed liquid chromatography–tandem mass spectrometry (LC-MS/MS) coupled with label-free quantitative proteomics. This approach allowed us to identify differentially expressed proteins (DEPs) in response to ammonium sulfate treatment compared to an untreated control. Subsequent bioinformatic analyses focused on characterizing secreted proteins and those associated with biocontrol functions. The primary objective of this study was to investigate how ammonium sulfate modulates the synthesis and secretion of key effector proteins in Czk1, thereby providing crucial proteome-level insights that can guide the rational design of more effective biocontrol formulations against rubber tree root diseases.

## 2. Materials and Methods

### 2.1. Extraction and Activity Detection of Antifungal Protein Crude Extract

The single colony of *B. subtilis* Czk1 was inoculated into sterile LB liquid medium and cultured at 28 °C with 180 rpm agitation for 12 h to prepare a seed culture. The seed culture was then transferred to fresh LB medium at 7% (*v*/*v*) inoculum density and incubated under identical conditions (28 °C, 180 rpm) for 60 h. Bacterial cells were subsequently removed by centrifugation at 12,000× *g* for 30 min at 4 °C. To specifically analyze the existing secreted proteins without further stimulating live bacteria, solid ammonium sulfate was directly added to the cell-free supernatant with continuous stirring until saturation. This approach allows efficient precipitation of already secreted proteins for proteomic analysis while avoiding potential changes in bacterial gene expression that could occur if ammonium sulfate were introduced during active fermentation. The mixture was maintained at 4 °C for 24 h to allow complete precipitation, followed by centrifugation at 12,000× *g* for 30 min at 4 °C. The resulting pellet containing crude proteins was resuspended in a minimal volume of phosphate buffer (pH 7.2) and subjected to dialysis for desalting. This crude protein extract was used for proteomic analysis.

### 2.2. Protein Extraction

Frozen samples were thawed on ice and transferred to sterile tubes. An appropriate amount of protein lysate (8 M urea, 1% SDS, containing protease inhibitors) was added. The samples were homogenized using a high-throughput tissue mill with three cycles of 40 s each time, followed by lysis on ice for 30 min with periodic vortexing (5–10 s every 5 min). The lysates were then centrifuged at 12,000× *g* for 30 min at 4 °C. The supernatant was collected, and its protein concentration was determined using the bicinchoninic acid (BCA) assay kit (Pierce, Thermo Scientific, Waltham, MA, USA) according to the manufacturer’s instructions. Briefly, a standard curve was prepared using bovine serum albumin (BSA) at concentrations ranging from 0 to 2.0 mg/mL. Each sample (2 µL) was diluted with 18 µL distilled water, mixed with 200 µL BCA working reagent, and incubated at 37 °C for 30 min with shaking. Absorbance was measured at 562 nm using a SPECTRA MAX microplate reader. Protein extraction quality was assessed by sodium dodecyl sulfate–polyacrylamide gel electrophoresis (SDS–PAGE) [33,34].

### 2.3. Protein Digestion and Peptide Desalting

A 100 μg protein aliquot was first denatured in 100 mM triethylammonium bicarbonate (TEAB) buffer. Reduction was performed with 10 mM tris(2-carboxyethyl)phosphine (TCEP) at 37 °C for 60 min, followed by alkylation using 40 mM iodoacetamide (IAM) for 40 min at 25 °C in the dark. Proteins were then precipitated with six volumes of pre-chilled acetone and incubated at −20 °C for 4 h. The pelleted material were collected by centrifugation at 10,000× *g* for 20 min at 4 °C. and subsequently resuspended in 100 μL of 100 mM TEAB. Trypsin was added at an enzyme-to-protein ratio of 1:50 (*w*/*w*) and digestion proceeded at 37 °C for 16 h.

The resulting peptides were dried under vacuum, reconstituted in 0.1% (*v*/*v*) trifluoroacetic acid (TFA), and desalted using HLB solid-phase extraction cartridges. After vacuum drying again, peptide concentration was determined with a quantitative peptide assay kit (Thermo Scientific, Cat. No. 23275). Finally, samples were prepared for LC–MS/MS analysis by dissolving dried peptides in loading buffer to a final concentration of 0.25 μg/μL [35].

### 2.4. LC-MS/MS Analysis

Tryptic peptides were analyzed using an EASY nLC-1200 system (Thermo Scientific, Waltham, MA, USA) coupled online to a Q Exactive HF-X quadrupole Orbitrap mass spectrometer (Thermo Scientific, Waltham, MA, USA) at Majorbio Bio-Pharm Technology Co. Ltd. (Shanghai, China). After being solubilized in mass spectrometry-compatible buffer, the peptides were loaded onto a C18 reversed-phase analytical column (75 μm × 25 cm, Thermo Scientific) equilibrated with solvent A (2% acetonitrile, 0.1% formic acid). Separation was performed using a linear gradient with solvent B (80% acetonitrile, 0.1% formic acid) at a flow rate of 300 nL/min under the following conditions: 0–45 min, 0% to 3% B; 45–50 min, 3% to 28% B; 50–55 min, 28% to 44% B; 55–60 min, 44% to 90% B.

Mass spectrometry acquisition was conducted in data-dependent mode. Full-scan MS spectra (*m*/*z* 350–1500) were acquired in the Orbitrap at a resolution of 60,000 with an automatic gain control (AGC) target of 3 × 10^6^ and a maximum injection time of 20 ms. The top 20 most intense precursor ions were selected for fragmentation via higher-energy collisional dissociation (HCD) with a normalized collision energy setting of 28%. MS/MS spectra were acquired at a resolution of 15,000 (at *m*/*z* 100), with an AGC target of 1 × 10^5^, a maximum injection time of 50 ms, and a dynamic exclusion duration of 18 s [36].

### 2.5. Protein Discovery Results

Raw MS data were processed using ProteomeDiscoverer™ Software (v 2.2, Thermo Scientific). Database searching was performed using MaxQuant [37] (v 2.0.3.1) with the following parameters: carbamidomethylation was designated as a fixed modification, while methionine oxidation was designated as a variable modification; trypsin was designated as the protease; the precursor mass tolerance was set to ±10 ppm; and the false discovery rate (FDR) for peptide identification was set to 1% (FDR ≤ 0.01). Only proteins containing at least one unique peptide were considered confidently identified.

To evaluate global sample relationships, principal component analysis (PCA) [38] was performed on the protein abundance matrix using the R programming environment (v 4.2.1). Abundance values were log_2_-transformed and scaled to unit variance prior to PCA. The analysis was conducted using the prcomp function, and the resulting scores were visualized in a two-dimensional plot to assess overall sample distribution and group clustering.

The protein abundance information obtained from the library search was used to analyze the statistical test for differential proteins. Student’s *t*-test was employed to calculate the significance of the difference between samples, and proteins exhibiting a fold change (FC) ≥ 1.2 or ≤0.83 and a *p*-value < 0.05 were designated as differentially expressed proteins (DEPs) [27,28].

### 2.6. Bioinformatics Analysis

#### 2.6.1. Functional Annotation and Enrichment Analysis

Functional annotation of the target proteins was performed using a combination of bioinformatics tools and databases [39]. KEGG [40] pathway annotations were assigned with KOBAS (v 2.1.1). Homologous gene clustering and functional classification were carried out using EggNOG (v 2020.06) [41]. Gene Ontology (GO) annotation was conducted with Blast2GO (v 2.5.0) [42] based on the Gene Ontology database (v 2022.0915). Protein domain analyses were performed using the Pfam database (v 35.0) [43]. Subcellular localization of differentially expressed proteins (DEPs) was predicted using ngloc (2023 version) by comparison against reference subcellular localization databases.

To identify functionally enriched terms and pathways, GO enrichment analysis was carried out with goatools (v 0.6.5) across the three major GO categories: biological process, cellular component, and molecular function. KEGG pathway enrichment analysis was performed using custom Python scripts (v 3.8.10).

#### 2.6.2. Metabolic Pathway Visualization

Global metabolic pathway mapping and visualization were conducted using iPath3.0 [44] to illustrate systemic metabolic networks and functional interactions among the annotated proteins.

## 3. Results

### 3.1. Extraction and Identification of Secreted Protein

The culture medium of *B. subtilis* Czk1 was debuffered, concentrated, and separated by SDS-PAGE (Figure 1). Electrophoretic analysis revealed that Czk1 secretes a diverse array of proteins.

### 3.2. Peptide Segment Information

Post-quality control statistical analysis of identified peptides included visualization of peptide count and length distributions. Figure 2 illustrates the number of peptides identified per protein, indicating robust proteome coverage.

### 3.3. Distribution of Protein Identified by B. subtilis Czk1

Venn diagram analysis (Figure 3) revealed distinct proteomic profiles between ammonium sulfate-treated *B. subtilis* Czk1 and the untreated control (CK). The Czk1 group expressed a total of 394 proteins, while the CK group expressed 106 proteins. Among these, 106 proteins were common to both groups. Notably, 288 proteins were exclusively identified in the Czk1 treatment group, suggesting a substantial alteration in the proteome in response to ammonium sulfate stress.

### 3.4. Sample Correlation Heat Map

Correlation analysis was performed to evaluate the similarity in protein composition among the biological replicates of the ammonium sulfate-treated *B. subtilis* Czk1 and the untreated control (CK) groups. As illustrated in Figure 4, high correlation coefficients (ranging from 0.996 to 0.997) were observed within the Czk1 replicates, indicating excellent reproducibility. Similarly, strong correlations (0.998–0.999) were detected within the CK group replicates. In contrast, the correlation coefficients between the Czk1 and CK groups were substantially lower (0.337–0.340), demonstrating pronounced divergence in their proteomic profiles and a marked treatment effect.

### 3.5. Principal Component Analysis

Principal Component Analysis (PCA) was performed to identify overall differences between sample groups, despite the limited sample size, as it effectively visualizes group segregation. Distinct sample clustering was observed in the PCA score plot (Figure 5), with clear separation between Czk1 and CK groups. All samples resided within the 95% confidence interval, confirming significant group segregation.

### 3.6. Analysis of Differentially Expressed Proteins

Differential expression analysis of proteins was conducted based on spectral processing and database searching. Differentially expressed proteins (DEPs) between *B. subtilis* Czk1 and the control (CK) group were identified using thresholds of |fold change| > 1.2 (up-regulated > 1.2, down-regulated < 0.83) and a significance level of *p* < 0.05. As shown in Figure 6, a total of 351 DEPs were identified, among which 329 were significantly up-regulated and 22 were down-regulated in the Czk1 group compared to CK. The marked predominance of up-regulated proteins suggests a robust proteomic response to ammonium sulfate treatment.

### 3.7. GO Function Annotation of Differentially Expressed Proteins

GO annotation analysis was performed to categorize the functional roles of DEPs identified in *B. subtilis* Czk1 under ammonium sulfate treatment, using a significance threshold of *p* < 0.05. GO annotation categorized DEPs into Biological Process (BP), Cellular Component (CC), and Molecular Function (MF) ontologies (Figure 7). In the BP category, Up-regulated DEPs were significantly enriched in a wide range of processes including biological regulation, metabolic process, reproduction, cellular process, developmental process, localization, signaling, response to stimulus. Conversely, down-regulated DEPs exhibited expression solely in the metabolic process and cellular process. Within the CC category, both up- and down-regulated DEPs were represented in protein-containing complexes and cellular anatomical entities. Within MF, down-regulated DEPs were exclusively associated with binding and catalytic activity. Up-regulated DEPs were expressed in a number of other activities, including translation regulator activity, transcription regulator activity, structural molecule activity, ATP-dependent activity, protein folding chaperone, antioxidant activity, transporter activity, small molecule sensor activity, and binding and catalytic activity.

These results indicate that ammonium sulfate treatment triggers extensive functional responses in *B. subtilis* Czk1, particularly enhancing regulatory, transport, antioxidant, and transcriptional activities.

### 3.8. KEGG Metabolic Pathway of Differentially Expressed Proteins

To elucidate the functional pathways involved in the response of *B. subtilis* Czk1 to ammonium sulfate treatment, KEGG pathway enrichment analysis was performed on DEPs. The DEPs were categorized into five classes: Cellular Processes, Human Diseases, Genetic Information Processing, Environmental Information Processing, and Metabolism (Figure 8). Within Cellular Processes, DEPs were associated with Cell motility and Cellular community—prokaryotes. Human Diseases has metabolic pathways was Drug resistance: antimicrobial. Genetic Information Processing comprised folding, sorting and degradation; replication and repair; and translation. Similarly, the field of Environmental Information Processing is characterized by two metabolic pathways: signal transduction and membrane transport. Metabolism encompasses 12 distinct metabolic pathways, including Global and overview maps, Carbohydrate metabolism, Amino acid metabolism, Metabolism of cofactors and vitamins, Lipid metabolism, Energy metabolism, Metabolism of other amino acids, Nucleotide metabolism, Biosynthesis of other secondary metabolites, Glycan biosynthesis and metabolism, Metabolism of terpenoids and polyketides, Xenobiotics biodegradation and metabolism.

The up-regulated DEPs were expressed in the aforementioned pathways, whereas the down-regulated DEPs only detected in a limited number of pathways, including folding, sorting and degradation (under Genetic Information Processing), signal transduction (under Environmental Information Processing), and amino acid metabolism, carbohydrate metabolism, energy metabolism, global and overview maps, glycan biosynthesis and metabolism, lipid metabolism, metabolism of cofactors and vitamins and metabolism of terpenoids and polyketides (under Metabolism).

These findings suggest that ammonium sulfate treatment extensively reprograms cellular metabolic and regulatory activities in *B. subtilis* Czk1, with a pronounced up-regulation of proteins involved in diverse biosynthetic, energy-yielding, and stress-responsive pathways.

### 3.9. EggNOG Annotation

EggNOG functional annotation revealed distinct patterns between up- and down-regulated differentially expressed proteins (DEPs). Among the 224 up-regulated DEPs, annotations spanned 17 functional categories (Figure 9), covering processes such as energy production and conversion (C), amino acid transport and metabolism (E), nucleotide transport and metabolism (F), carbohydrate transport and metabolism (G), coenzyme transport and metabolism (H), lipid transport and metabolism (I), translation, ribosomal structure and biogenesis (J), transcription (K), replication, recombination and repair (L), cell wall/membrane/envelope biogenesis (M), cell motility (N), posttranslational modification, protein turnover, and chaperones (O), inorganic ion transport and metabolism (P), secondary metabolites biosynthesis, transport and catabolism (Q), signal transduction mechanisms (T), intracellular trafficking, secretion, and vesicular transport (U), and defense mechanisms (V).

The most enriched categories included amino acid transport and metabolism (E, 38 proteins), energy production and conversion (C, 32 proteins), carbohydrate transport and metabolism (G, 29 proteins), and cell wall/membrane/envelope biogenesis (M, 23 proteins). This enrichment pattern suggests enhanced biosynthesis of protein precursors, upregulated energy metabolism, altered glycolytic pathways, and active cellular remodeling, potentially related to adaptation or defense.

In contrast, the 15 down-regulated DEPs were mapped to only six functional categories, which were predominantly associated with core metabolic processes: energy production and conversion (C), amino acid transport and metabolism (E), nucleotide transport and metabolism (F), carbohydrate transport and metabolism (G), coenzyme transport and metabolism (H), and lipid transport and metabolism (I).

### 3.10. Pfam Classification

The DEPs were analyzed against the Pfam database to characterize their functional domains (Figure 10). Among the up-regulated proteins, 82 were annotated to 20 distinct functional domains, including amidohydro_1, amidase_2, alpha-amylase, aldo-ket-red, aldedh, AP-endonus, peptidase_M20, peptidase_M42, lactamase_B, Glu_dehyd C, Aminotran-1-2, ADH-zinc-N-2, adh-short-C2, peptidase-S8, nitroreductase, KR, ADH-zinc-N, ADH-N, glyoxalase, and adh-short. In contrast, only two down-regulated proteins were annotated, corresponding to two functional domains: peptidase_M20 and Glyoxalase.

The identified domains are predominantly categorized as oxidoreductases, which catalyze essential redox reactions, and hydrolases, which play a pivotal role in cleaving peptide, glycosidic, and amide bonds. Furthermore, key biosynthetic and metabolic functions are underscored by domains such as aminotransferases, beta-ketoacyl synthases, and the CoA-ligase/AMP-binding enzyme family. Collectively, these functional capabilities underpin primary metabolic processes, maintain cell envelope integrity, and facilitate critical environmental interactions.

### 3.11. Subloc Annotation

The subcellular localization information of DEPs were obtained by comparing the subcellular localization database, and its annotations in the database were statistically analyzed. As shown in Figure 11, the Czk1 DEPs were divided into three subcellular compartments: Extracellular (EXT), plasma membrane (PLA), and cytoplasm (CYT). The results of this analysis revealed that there were 13 up-regulated proteins in the EXT, 10 up-regulated proteins in the PLA, and 306 up-regulated proteins in the CYT. Conversely, 22 down-regulated DEPs were localized exclusively to PLA.

Quantitative analysis revealed a distinct distribution pattern among up-regulated DEPs. The majority (306 proteins) were localized to the CYT, suggesting a pronounced activation of intracellular metabolic or signaling processes. Additionally, 13 and 10 up-regulated proteins were identified in the EXT and PLA compartments, respectively, indicating potential modifications in secretory functions or membrane-related activities. In contrast, down-regulated DEPs exhibited a strikingly specific localization pattern, with all 22 significantly decreased proteins residing exclusively in the PLA. This strong bias implies a targeted suppression of pathways associated with membrane integrity, nutrient transport, cell adhesion, or signal transduction.

### 3.12. GO Enrichment

The GO function enrichment analysis of the DEPs were performed to clarify the biological processes involved in the protein, the cell components, and the molecular functions achieved at the functional level, and to clarify the main GO functions of the protein (Figure 12). For the up-regulated DEPs, molecular function terms were prominently enriched. Specifically, 36 proteins were associated with transferase activity, and 23 proteins were implicated in nucleic acid binding. Among the down-regulated DEPs, notable enrichment was observed in specific biological processes and molecular functions. In biological processes, three proteins were assigned to small molecule catabolic process and two to organic acid catabolic process. In molecular functions, 2 proteins each were enriched in the following terms: oxidoreductase activity, acting on the CH-CH group of donors, with a flavin as acceptor, acyl-CoA dehydrogenase activity,4 iron, 4 sulfur cluster binding and oxidoreductase activity, acting on the CH-CH group of donors. Additionally, 3 proteins were enriched each in iron-sulfur cluster binding, metal cluster binding, hydro-lyase activity and carbon-oxygen lyase activity.

Significant enrichment was identified for eight functional terms: small molecule catabolic process, oxidoreductase activity, acting on the CH-CH group of donors, with a flavin as acceptor, acyl-CoA dehydrogenase activity, iron-sulfur cluster binding, metal cluster binding, hydro-lyase activity,4 iron, 4 sulfur cluster binding, carbon-oxygen lyase activity.

### 3.13. KEGG Pathway Enrichment

KEGG pathway enrichment analysis revealed distinct metabolic signatures for DEPs (Figure 13). The up-regulated DEPs were mainly enriched in Glyoxylate and dicarboxylate metabolism, Arginine and proline metabolism, Biosynthesis of cofactors, Biosynthesis of nucleotide sugars, Biotin metabolism, Butanoate metabolism, Chloroalkane and chloroalkene degradation, Cysteine and methionine metabolism, Fatty acid biosynthesis, Fatty acid degradation, Fatty acid metabolism, Glycine, serine and threonine metabolism, Pantothenate and CoA biosynthesis, Porphyrin metabolism, Riboflavin metabolism, Sulfur metabolism, Tyrosine metabolism, Ubiquinone and other terpenoid-quinone biosynthesis, Vitamin B6 metabolism and beta-Alanine metabolism. The down-regulated DEPs showed significant enrichment in Thiamine metabolism, Microbial metabolism in diverse environments, Lysine biosynthesis, 2-Oxocarboxylic acid metabolism, Metabolic pathways, Biosynthesis of amino acids, Monobactam biosynthesis, Sulfur relay system, Glutathione metabolism, Peptidoglycan biosynthesis, Terpenoid backbone biosynthesis, Pentose phosphate pathway, C5-Branched dibasic acid metabolism, Citrate cycle (TCA cycle), Arginine biosynthesis, Histidine metabolism, Two-component system, Glyoxylate and dicarboxylate metabolism, Biosynthesis of secondary metabolites and Carbon metabolism.

These results indicate a broad rewiring of metabolic activity, with up-regulated DEPs enhancing energy production and anabolic processes, while down-regulated DEPs reflect a reduction in basal metabolic and biosynthetic functions.

## 4. Discussion

The future trajectory of sustainable agriculture will be increasingly shaped by the adoption of biological control strategies for plant disease management. *B. subtilis* is a prominent biocontrol bacterium, valued for its capacity to produce diverse antimicrobial proteins (AMPs). These metabolites offer advantages such as chemical stability, sustained efficacy, and broad-spectrum activity, conferring significant application potential. A deeper understanding of the antimicrobial proteome of *B. subtilis* is essential for elucidating its biocontrol mechanisms, improving its efficacy, and informing novel disease management strategies.

Recent advances in proteomics technologies, synergized with expanding genomic databases, have precipitated proteomics analysis to the forefront of scientific inquiry as a highly efficacious means to explore protein expression levels, protein modifications, and protein–protein interactions in diverse biological systems [45]. Among emerging techniques, label-free proteomics has emerged as a robust LC-MS/MS methodology that enables comparative protein quantification through spectral analysis of enzymatically digested peptides, eliminating the need for isotopic [46,47]. This technique provides an unbiased platform for systematic protein profiling under diverse physiological conditions. Previous studies have successfully applied label-free proteomics in diverse contexts: Wang et al. (2018) employed this approach to detect alterations in protein levels in *B. cereus* spores treated with HPP-SAEW [48]. Panda et al. (2020) conducted label-free quantitative proteomic analysis on shoots of *Haloxylon salicornicum* to unravel functional protein networks associated with salt tolerance [49]. Yang et al. (2022) employed label-free proteomics to elucidate the impact of harvest time on photosynthetic apparatus quality in the red alga *Porphyra dentata* [50]. Zhao et al. (2023) conducted comparative label-free quantitative analysis of the milk fat globule membrane profiles in porcine colostrum versus mature milk [51]. In this study, we implemented label-free quantitative proteomics to investigate ammonium sulfate-induced dynamics of the secretory protein in *B. subtilis* Czk1. A total of 351 DEPs were identified between treated and control groups. Bioinformatic analyses (KEGG and GO enrichment) revealed significant reprogramming of metabolic pathways closely linked to biocontrol functions. Crucially, ammonium sulfate induction enhanced the expression of proteins involved in antagonistic activities, including secondary metabolite synthesis, amino acid metabolism, and cofactor biosynthesis, while repressing pathways related to primary catabolism. This shift suggests a metabolic reallocation from growth to defense-related protein secretion, providing a proteomic basis for the enhanced antifungal effects observed.

Proteins in living systems operate within coordinated networks of biochemical pathways to execute biological functions. Pathway analysis is therefore essential for a holistic and systematic interpretation of cellular processes. Recent proteomic studies underscore its utility across biological models. For example, Jia et al. (2023) used data-independent acquisition (DIA) proteomics to show that energy transduction pathways were downregulated in *Bacillus cereus* spores under combined stress, suggesting a metabolic slowdown that promotes survival [52]. Wang et al. (2024) integrated HPLC-MS/MS with label-free proteomics to identify DEPs in *Lactobacillus plantarum* after bacteriocin exposure, noting enrichment in peptidoglycan catabolism, carbohydrate metabolism, and nucleotide biosynthesis [53]. In Czk1, we observed notable enrichment in oxidoreductase activities and small molecule catabolism, which are known to contribute to plant resistance and pathogen inhibition.

KEGG analysis revealed that the up-regulated DEPs participated in 95 pathways, while down-regulated DEPs participated in 29. Key enriched pathways included amino acid metabolism (glyoxylate/dicarboxylate, arginine/proline, cysteine/methionine, glycine/serine/threonine), cofactor biosynthesis (biotin, riboflavin, pantothenate/CoA, porphyrin, ubiquinone, vitamin B6), carbohydrate/fatty acid metabolism, and secondary metabolite biosynthesis. This reprogramming suggests that ammonium sulfate induction shifts central carbon and nitrogen metabolism in Czk1 toward the synthesis and secretion of antimicrobial compounds, such as secondary metabolites, amino acid derivatives, and enzyme cofactors. Concurrent down-regulation of core catabolic pathways—including the TCA cycle, amino acid biosynthesis, and carbohydrate metabolism—further indicates a metabolic transition from primary growth functions toward specialized metabolite production.

In conclusion, our integrated proteomic and bioinformatic analyses reveal that ammonium sulfate induction promotes a metabolic reconfiguration in *B. subtilis* Czk1, favoring the production and secretion of antimicrobial compounds over growth-associated processes. These insights not only clarify the mechanistic basis for improved biocontrol performance but also offer actionable molecular targets for designing effective *Bacillus*-based biofungicides. Future work should validate the functional roles of key DEPs and assess their efficacy in field applications to translate these proteomic discoveries into practical disease management solutions.

## 5. Conclusions

This study employed label-free quantitative proteomics to analyze ammonium sulfate-induced alterations in the secretory proteome of the biocontrol agent *Bacillus subtilis* Czk1. A total of 351 differentially expressed proteins (DEPs) were identified, including 329 up-regulated and 22 down-regulated proteins. GO enrichment analysis highlighted eight significantly enriched functional terms, largely associated with metabolic enzymes and catabolic processes. KEGG analysis assigned the DEPs to 124 distinct pathways, with significant enrichment in pathways central to amino acid metabolism, cofactor biosynthesis, glyoxylate/dicarboxylate metabolism, and secondary metabolite production. Crucially, 13 up-regulated proteins were predicted to be extracellular, directly implicating them in the secreted biocontrol arsenal. This work establishes a robust label-free quantitative-based workflow for secretome profiling in biocontrol bacteria. Our results demonstrate that ammonium sulfate induction markedly reprograms the secretome of strain Czk1, promoting the expression of key proteins involved in antimicrobial metabolic pathways. These findings provide crucial molecular targets for understanding and enhancing the biocontrol mechanisms of *B. subtilis* Czk1, facilitating the development of more effective biofungicides against economically devastating rubber tree root rot diseases caused by *Ganoderma pseudoferreum* and *Pyrrhoderma noxium*. Future work should focus on validating the antifungal activity of the identified secretory proteins and optimizing fermentation conditions to maximize their production for large-scale biocontrol applications.

## Figures and Tables

**Figure 1 microorganisms-13-02212-f001:**
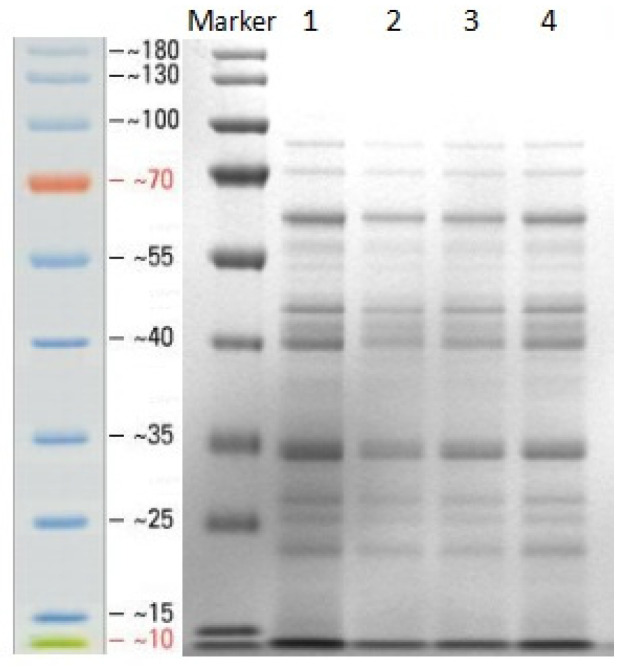
SDS-PAGE analysis of secreted proteins from *B. subtilis* Czk1 culture supernatant. (Lane M, protein marker (10–180 kDa); Lane 1–2: Czk1 sample protein, Lane 3–4: CK control sample protein).

**Figure 2 microorganisms-13-02212-f002:**
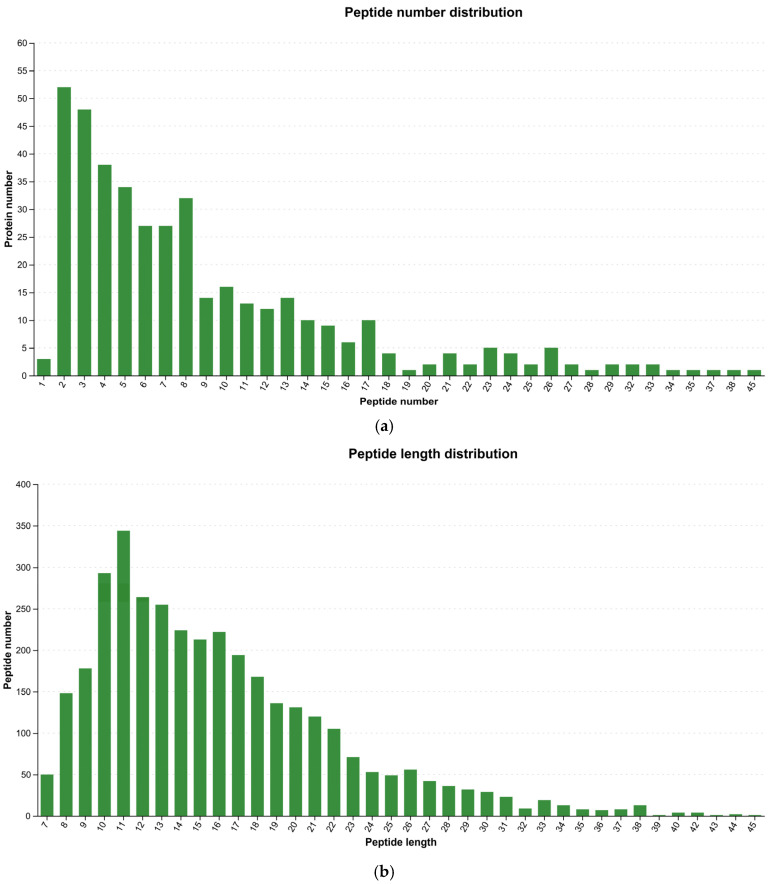
Characterization of the identified peptides. (**a**) Distribution of the number of peptides per protein. (**b**) Distribution of peptide lengths.

**Figure 3 microorganisms-13-02212-f003:**
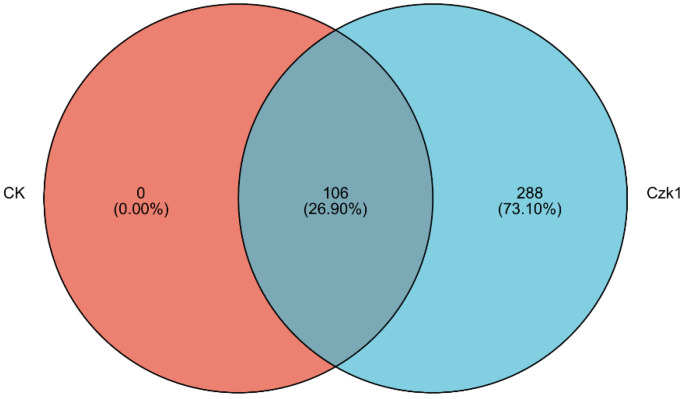
Venn diagram of proteins identified in the *B. subtilis* Czk1 group versus the control (CK) group. Note: The overlapping part of the figure represents the number of proteins shared between the groups, the non-overlapping part represents the number of proteins unique to the group, and the number represents the number of corresponding proteins.

**Figure 4 microorganisms-13-02212-f004:**
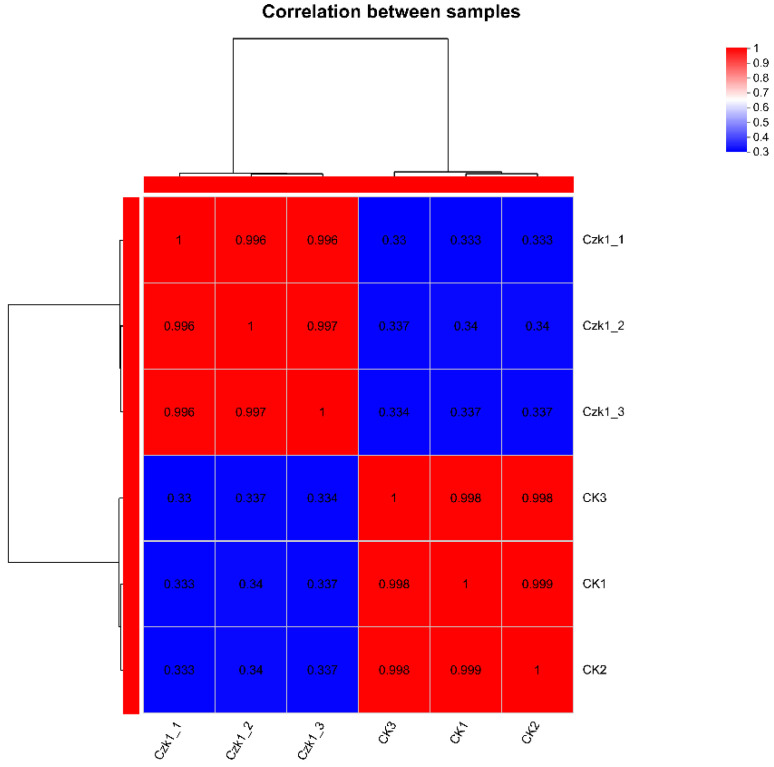
Heatmap of sample correlation coefficients for the Czk1 group and CK group. Note: The right and lower sides of the figure are the sample names. Each grid in the graph represents the correlation between two samples, and different colors represent the size of the correlation coefficient between samples.

**Figure 5 microorganisms-13-02212-f005:**
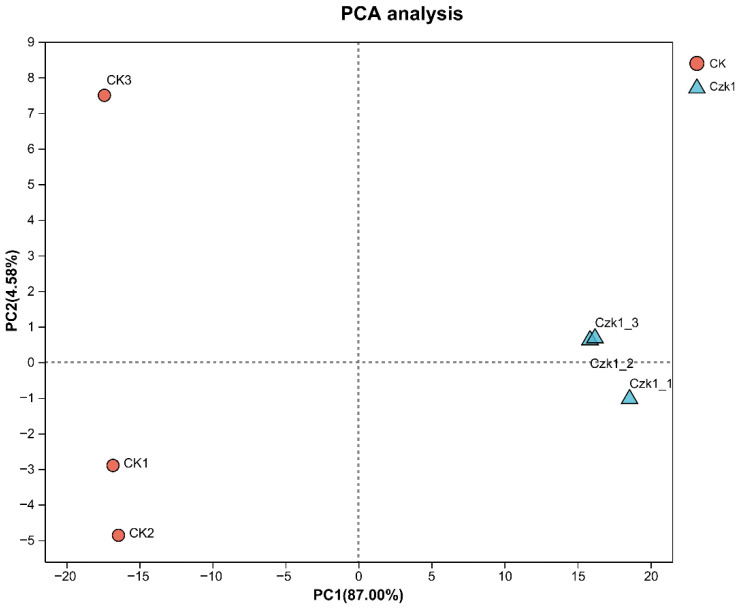
PCA score scatter plot of Czk1 group and CK group. Note: Following the completion of the dimension reduction analysis, the relative coordinate points on the principal component plane are identified. The distance between each sample point represents the degree of similarity between the samples. A shorter distance indicates a higher degree of similarity between the samples.

**Figure 6 microorganisms-13-02212-f006:**
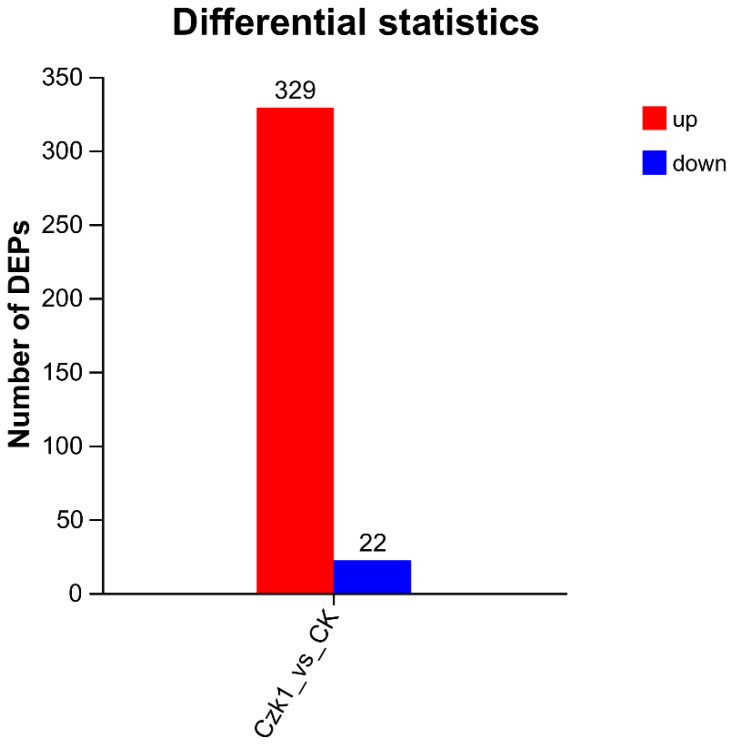
Summary of differentially expressed proteins (DEPs). Note: The abscissa represents the difference between the comparison groups. The default red color represents the protein that has been shown to be up-regulated in the different comparison group, while the blue color represents the protein that has been demonstrated to be down-regulated.

**Figure 7 microorganisms-13-02212-f007:**
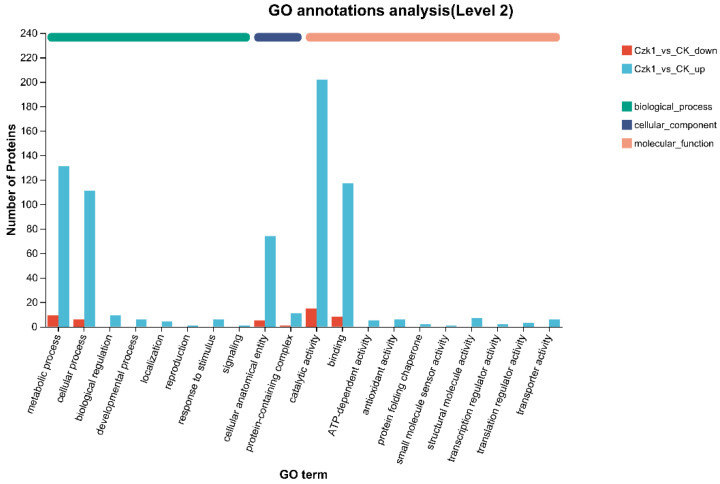
GO functional annotation analysis of DEPs. Note: Each column in the figure represents a multi-level classification of GO, and the higher the column, the more proteins in this multi-level classification; the abscissa represents the multi-level classification term of GO, and the ordinate represents the number of differential proteins annotated to the multi-level classification.

**Figure 8 microorganisms-13-02212-f008:**
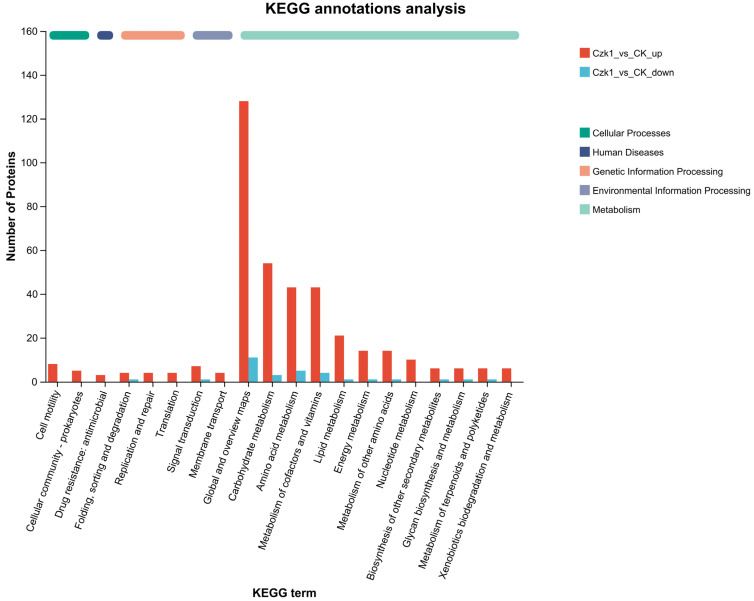
KEGG pathway annotation analysis of DEPs. Note: The ordinate in the figure is the name of the KEGG metabolic pathway, and the abscissa is the number of proteins annotated to the pathway. Paths of different colors represent different categories to which they belong.

**Figure 9 microorganisms-13-02212-f009:**
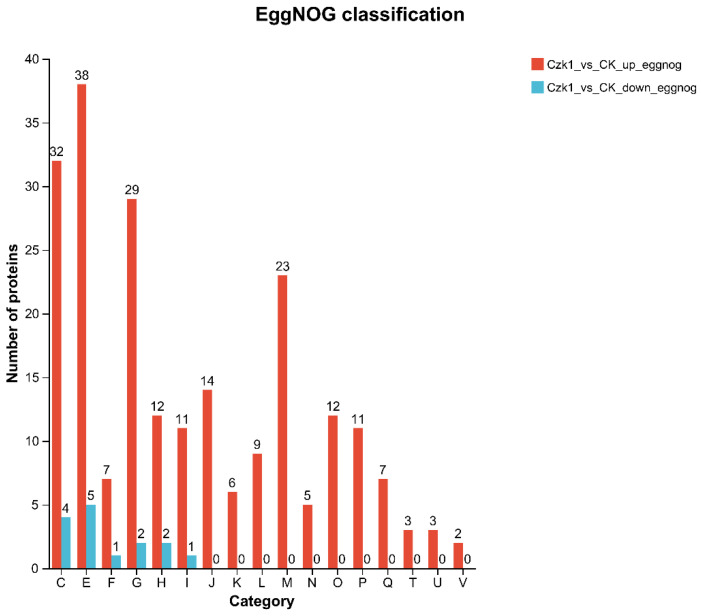
EggNOG functional classification of DEPs. Note: The abscissa represents the functional classification of EggNOG (represented by capital letters C–V), and the ordinate represents the number of proteins with this function. (C) Energy production and conversion; (E) Amino acid transport and metabolism; (F) Nucleotide transport and metabolism; (G) Carbohydrate transport and metabolism; (H) Coenzyme transport and metabolism; (I) Lipid transport and metabolism; (J) Translation, ribosomal structure and biogenesis; (K) Transcription; (L) Replication, recombination and repair; (M) Cell wall/membrane/envelope biogenesis; (N) Cell motility; (O) Posttranslational modification, protein turnover, chaperones; (P) Inorganic ion transport and metabolism; (Q) Secondary metabolites biosynthesis, transport and catabolism; (T) Signal transduction mechanisms; (U) Intracellular trafficking, secretion, and vesicular transport; (V) Defense mechanisms.

**Figure 10 microorganisms-13-02212-f010:**
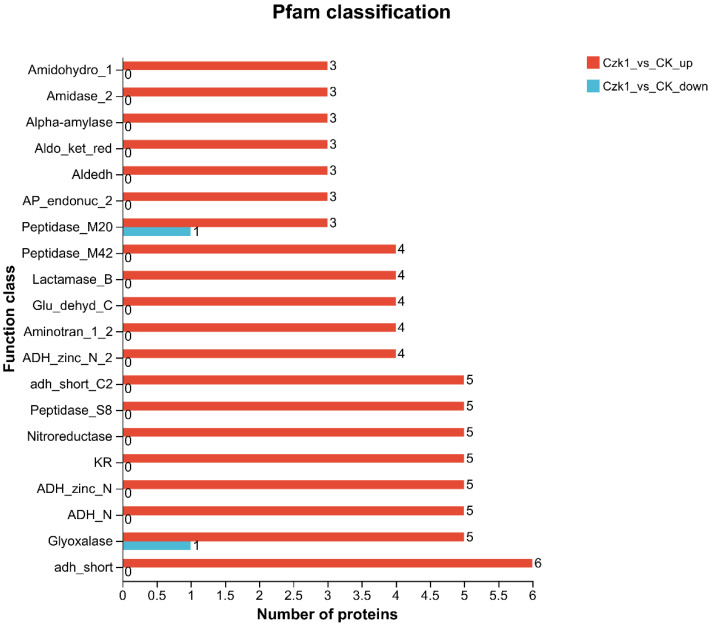
Pfam domain classification of DEPs. Note: The ordinate is the abbreviation of the functional domain (domain), and the abscissa is the number of proteins annotated to each domain.

**Figure 11 microorganisms-13-02212-f011:**
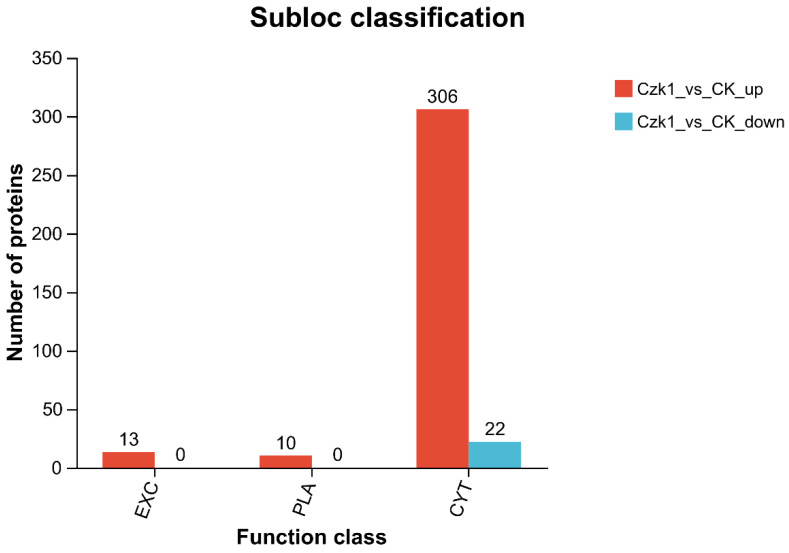
Subcellular localization prediction of DEPs. Note: The abscissa is the name of different subcellular organelles, and the ordinate is the number of proteins annotated to the subcellular organelles. Localization class and corresponding code are CYT: Cytoplasm, PLA: Plasma Membrane, EXC: Extracellular.

**Figure 12 microorganisms-13-02212-f012:**
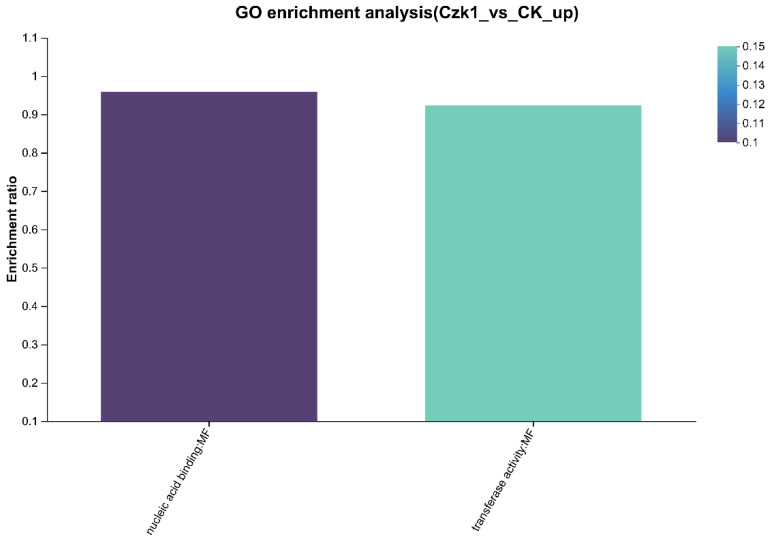
GO enrichment analysis of DEPs. Note: The abscissa represents the GO term, and the ordinate represents the enrichment rate (calculated as the number of DEPs enriched in the term divided by the number of background proteins annotated to that term). A higher value indicates greater enrichment. The column color gradient represents the significance of the enrichment, where *p* or FDR < 0.05 is marked as *.

**Figure 13 microorganisms-13-02212-f013:**
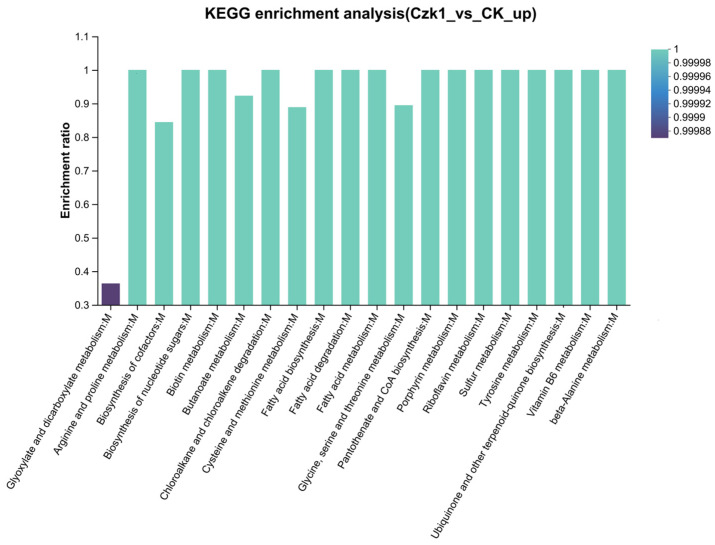
KEGG pathway enrichment analysis of DEPs.

## Data Availability

The datasets presented in this article are not readily available due to privacy reasons. Requests to access the datasets should be directed to corresponding authors.

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
