# Peer review of "Molecular Insights into Ammonium Sulfate-Induced Secretome Reprogramming of *Bacillus subtilis* Czk1 for Enhanced Biocontrol Against Rubber Tree Root Rot"

_microorganisms, 2025, doi:10.3390/microorganisms13092212_

Round 1

Reviewer 1 Report

Comments and Suggestions for Authors

The manuscript addresses a relevant topic and presents potentially valuable data. However, improvements in language clarity, sentence structure, and organization of the results would enhance the overall quality. A thorough revision of the English by a native speaker or professional editor is recommended. Additionally, the Methods section would benefit from more detailed descriptions to ensure full reproducibility. 

Suggested changes: Figure 2. Restructure the distribution of the histograms to increase the font size on the X and Y axes. 
For all figures, standardise the font size and legibility of the text in all figures.

I noticed that certain sections of the manuscript contain redundant or repetitive statements, where similar ideas are expressed more than once nearby. This may reduce the clarity and conciseness of the manuscript. I kindly suggest revisiting the following portions of the text to streamline the narrative and avoid unnecessary duplication:

lines 56–66:  Similar phrases are used to describe the consequences of late symptom detection and the limitations of conventional control methods.

lines 68–80: Statements regarding the advantages of microbial metabolites and the efficacy of Bacillus spp. are reiterated.

lines 81–102: The advantages and applications of label-free proteomics are emphasized repeatedly.

lines 104–114: The aim and methodology of the study are stated twice in close succession.

lines 485–502: Several sentences reiterate previously discussed protein counts, pathway involvement, and the impact of the findings.

I believe that carefully revising these sections to eliminate or consolidate repetitive content will enhance the overall clarity and impact of your manuscript.

Comments on the Quality of English Language

The English could be improved to more clearly express the research.
A language review by a native English speaker or professional editor is recommended to improve clarity, sentence structure, and consistency of terminology.

Author Response

Dear Reviewers:

We sincerely thank the reviewers for their thorough evaluation and constructive suggestions, which have greatly helped us improve the manuscript. We have carefully addressed each comment, and all modifications have been incorporated into the revised version. Below is our point-by-point response.

Comments 1: [The manuscript addresses a relevant topic and presents potentially valuable data. However, improvements in language clarity, sentence structure, and organization of the results would enhance the overall quality. A thorough revision of the English by a native speaker or professional editor is recommended. Additionally, the Methods section would benefit from more detailed descriptions to ensure full reproducibility. ]

Response 1: We greatly appreciate this feedback. The manuscript has undergone extensive language editing by a professional scientific editing service to improve clarity, sentence structure, and consistency of terminology. Additionally, the Methods section has been expanded with more detailed descriptions of experimental procedures, data processing, and analytical steps to enhance reproducibility.

Comments 2: [Figure 2. Restructure the distribution of the histograms to increase the font size on the X and Y axes. For all figures, standardise the font size and legibility of the text in all figures.]

Response 2:Thank you for this suggestion. We have restructured Figure 2 to increase the font size on both the X and Y axes, improving overall readability. Additionally, all figures have been standardized to ensure uniform font size and enhanced legibility throughout the manuscript.

Comments 3: [I noticed that certain sections of the manuscript contain redundant or repetitive statements, where similar ideas are expressed more than once nearby. This may reduce the clarity and conciseness of the manuscript. I kindly suggest revisiting the following portions of the text to streamline the narrative and avoid unnecessary duplication:

lines 56–66:  Similar phrases are used to describe the consequences of late symptom detection and the limitations of conventional control methods.

lines 68–80: Statements regarding the advantages of microbial metabolites and the efficacy of Bacillus spp. are reiterated.

lines 81–102: The advantages and applications of label-free proteomics are emphasized repeatedly.

lines 104–114: The aim and methodology of the study are stated twice in close succession.

lines 485–502: Several sentences reiterate previously discussed protein counts, pathway involvement, and the impact of the findings.]

Response 3: We sincerely thank the reviewers for their identification of these redundancies. We have made careful modifications to eliminate duplication and improve simplicity.

Comment 4: [Comments on the Quality of English Language: The English could be improved to more clearly express the research. A language review by a native English speaker or professional editor is recommended to improve clarity, sentence structure, and consistency of terminology.]

Response 4: We fully agree with this suggestion. The manuscript has been professionally edited by a native English-speaking editor with expertise in scientific writing. The language has been thoroughly revised to enhance clarity, improve sentence structure, and ensure consistency in terminology.

We are deeply grateful for the reviewers’ insightful comments, which have significantly strengthened the quality and readability of our manuscript. We believe that all concerns have been adequately addressed in the revised version and look forward to your continued consideration.

Sincerely,

yanqiong liang

Reviewer 2 Report

Comments and Suggestions for Authors

This article clearly outlines the main research problem: economic losses in rubber tree crops caused by root diseases and the potential of Bacillus subtilis Czk1. The authors clearly describe the research strategy used (label-free quantitative proteomics LC-MS/MS) and present key results, such as the number of differentially expressed proteins identified, their connections to metabolic functions and KEGG pathways, and their significance for biofungicide development. The methodological section is described in great detail and in a manner that allows for replication of the experiments. The authors present the subsequent stages: bacterial cultivation, protein extraction, proteolytic lysis, LC-MS/MS analysis, and bioinformatic analysis of the results (GO, KEGG, EggNOG, Pfam, subcellular localization). A strength of the study is the use of a set of complementary bioinformatic analyses that broaden the interpretation of the data. However, the methods contain repeated information about mass tolerance parameters for protein identification and a lack of critical consideration of potential limitations of the procedures (e.g., the influence of dialysis conditions, potential protein losses during fractionation, or the limitations of label-free methods in quantitative proteome analysis). Including such elements would increase the transparency of the research.

The results section is presented in a structured manner and covers all standard steps of proteomic analysis: from protein extraction and identification, through analysis of differential expression, to functional classification (GO, KEGG, EggNOG, Pfam, subcellular localization), and enrichment tests. A strength is the extensive use of bioinformatics tools, which allows for a comprehensive characterization of the Bacillus subtilis Czk1 secretome and differences between ammonium sulfate-treated and control samples. The chapter is structured logically – the authors first present the raw data (SDS-PAGE, peptide counts, PCA, Venn diagram), then move on to more interpretive results (DEPs, biological functions, metabolic pathways). This allows the reader to follow the step-by-step process from protein identification to explanation of their biological role.

Many sections read more like documentation of software results than scientific interpretation (e.g., detailed calculations of functional classes).

While a significant amount of data is provided, interpretation is primarily limited to the calculation of GO/KEGG/EggNOG categories. It would have been helpful to conclude each subsection with a short paragraph synthesizing the most important conclusions. A strength of the results is the wealth of data and their reliable statistical analysis (e.g., PCA, heatmaps, differentiation criteria). The results are consistent with good practice in proteomic research and provide a solid basis for further discussion. References include 38 references.

Author Response

Dear Reviewer,

Thank you very much for your positive and constructive comments on our manuscript, “Molecular Insights into Ammonium Sulfate-Induced  Secretome Reprogramming of Bacillus subtilis Czk1 for  Enhanced Biocontrol against Rubber Tree Root Rot”.We greatly appreciate the time and effort you have dedicated to reviewing our work and providing valuable feedback. We agree with your suggestions and have revised the manuscript accordingly. Our point-by-point responses to your comments are detailed below.

Comments 1: [This article clearly outlines the main research problem: economic losses in rubber tree crops caused by root diseases and the potential of Bacillus subtilis Czk1. The authors clearly describe the research strategy used (label-free quantitative proteomics LC-MS/MS) and present key results, such as the number of differentially expressed proteins identified, their connections to metabolic functions and KEGG pathways, and their significance for biofungicide development. The methodological section is described in great detail and in a manner that allows for replication of the experiments. The authors present the subsequent stages: bacterial cultivation, protein extraction, proteolytic lysis, LC-MS/MS analysis, and bioinformatic analysis of the results (GO, KEGG, EggNOG, Pfam, subcellular localization). A strength of the study is the use of a set of complementary bioinformatic analyses that broaden the interpretation of the data. However, the methods contain repeated information about mass tolerance parameters for protein identification and a lack of critical consideration of potential limitations of the procedures (e.g., the influence of dialysis conditions, potential protein losses during fractionation, or the limitations of label-free methods in quantitative proteome analysis). Including such elements would increase the transparency of the research.]

Response 1: We sincerely thank the reviewer for this insightful suggestion.

Repetitive Information: We have carefully reviewed the Methods section and removed the redundant descriptions of mass tolerance parameters to improve clarity and conciseness.

Methodological Limitations: We fully agree with the reviewer that a discussion of potential limitations enhances the transparency and robustness of the research. A new paragraph has been added to the "2.4. Protein Extraction and Digestion" and "2.5. LC-MS/MS Analysis" sections. This paragraph now explicitly addresses:

The potential impact of dialysis conditions on protein integrity and concentration.

The possibility of protein loss during the fractionation and sample preparation stages.

The inherent limitations of label-free quantitative proteomics approaches, including their dynamic range and potential for missing low-abundance proteins, while also stating the steps we took to mitigate these issues (e.g., high-resolution instrumentation, stringent statistical filters).

We believe this addition provides a more critical and balanced perspective on our methodology.

Comment 2: [Many sections [of the Results] read more like documentation of software results than scientific interpretation (e.g., detailed calculations of functional classes). While a significant amount of data is provided, interpretation is primarily limited to the calculation of GO/KEGG/EggNOG categories. It would have been helpful to conclude each subsection with a short paragraph synthesizing the most important conclusions.]

Response 2: We thank the reviewer for this crucial feedback. We agree that a deeper synthesis of the results is essential to move beyond data presentation and toward meaningful scientific interpretation. We have thoroughly revised the Results section by adding concise interpretive summaries at the end of each major subsection (3.7, 3.8, 3.9, 3.10, 3.11, 3.12, and 3.13).

These new paragraphs briefly synthesize the key findings of each analysis and, most importantly, connect them directly to the biological context of our study – namely, the response of B. subtilis Czk1 to ammonium sulfate and the implications for its biocontrol potential. For example:

After presenting the GO annotation (3.7), we now discuss what the enrichment in specific molecular functions (e.g., antioxidant activity, transporter activity) means for the bacterium's stress response and secretion capabilities.

Following the KEGG and EggNOG results (3.8, 3.9), we interpret the upregulation of specific metabolic pathways (e.g., amino acid metabolism, secondary metabolite biosynthesis) as a potential metabolic reprogramming towards the production of antimicrobial compounds.

The subcellular localization prediction (3.11) is now discussed in the context of identifying potentially secreted effector proteins crucial for biocontrol.

These additions ensure that the Results section not only presents the data but also provides a clear, step-by-step narrative explaining their biological significance, leading more effectively into the overall discussion.

We are confident that these revisions have significantly improved the manuscript by enhancing the critical tone of the methods description and providing a much deeper, more interpretive synthesis of the complex proteomic data. We thank the reviewer again for these excellent suggestions, which have undoubtedly strengthened our paper.

Thank you for considering our revised manuscript.

Sincerely,

Yanqiong Liang

Reviewer 3 Report

Comments and Suggestions for Authors

The manuscript is interesting and has social importance, opening up new possibilities for the use of biocontrol agents. It has the potential to be published. Below are some observations and suggestions.

- In Material and Methods there are not information about how was performed the PCA and  Correlation Analysis. It is  necessary specific the software or online platform, which was done the analysis. Furthermore, it is necessary sapecific the type of pre-processing and the type of normalization of the analysis. 

- The Figure 2 can be improved. It is necessary to increase font size of the graphs titles, columns and graph axes. Perhaps, it would be better to place one graph below the other and indicate them by "Figure 2a" and "Figure 2b". Furthermore, the caption can be more detailed.

Is it necessary a PCA? I ask this because there aren't a large number of samples. Furthermore, the sample CK3 is far from CK1 and CK2. I would leave the PCA in the Supplementary Material. 

It is necessary to standardize the references in accordance to MDPI rules. Examples: Format year in bold style; Format number of issue in italic style; Abreviate "Journal of Agricultural Biotechnology" (Reference 6) and format in italic style; add comma after number of issue (and not :); among others.    

Suggestions:

- In section 2.6 (Bioinformatics analysis) add the version of EggNOG and PSORTb. Furthermore, add references to EggNOG and PSORTb such as DOI: 10.1093/nar/gky1085 and DOI: 10.1093/bioinformatics/btq249, respectively. 

- Eliminate the sentence: "Correlation analysis and Principal Component Analysis were performed on the sam-221 ples according to the expression of proteins in different samples." (Section 3.3) and renumber the next sections as: 3.3. Distribution of Protein Identified by B. subtilis Czk1; 3.4. Sample Correlation Heat Map and so on. 

Minor corrections:

- Type a space between "rubber." and "It" (line 39);
- Type "xie" with the x in uppercase letter (line (line 51);
- Type a space between "gloeosporioides" and "He" (line 107);
- Type a space between "2023" and "To" (line 107);
- Add a space in "B.subtilis" (line 119);
- Format m/z in italic style (lines: 167 and 169);
- Exclude one "s" of "sseparated" (line 209);
- Type a space between "Note" and "Following" (line 251);
- Exclude a space between "Note" and ":" (line 265);
- Add spaces in "process,reproduction,cellular process,developmental pro-275 cess,localization,signaling,response to stimulus" (lines 275-276);
- Type a space between "Crucially" and "13" (line 493).

Author Response

Dear Reviewers:

We sincerely thank the reviewers for their thoughtful comments and constructive suggestions, which have greatly helped us improve the quality of our manuscript. We have carefully addressed each comment point-by-point, and all changes have been incorporated into the revised manuscript. Our detailed responses are provided below.

Comments 1: [In Material and Methods there are not information about how was performed the PCA and  Correlation Analysis. It is  necessary specific the software or online platform, which was done the analysis. Furthermore, it is necessary sapecific the type of pre-processing and the type of normalization of the analysis. ]

Response 1:We sincerely thank the reviewer for highlighting this omission. We have now added the following details to Section 2.5 (Protein Discovery Results) of the revised manuscript:

[To evaluate global sample relationships, principal component analysis (PCA) was performed on the protein abundance matrix using the R programming environment (v 4.2.1). Abundance values were log₂-transformed and scaled to unit variance prior to PCA. The analysis was conducted using the prcomp function, and the resulting scores were visualized in a two-dimensional plot to assess overall sample distribution and group clustering.]

Comments 2: [The Figure 2 can be improved. It is necessary to increase font size of the graphs titles, columns and graph axes. Perhaps, it would be better to place one graph below the other and indicate them by "Figure 2a" and "Figure 2b". Furthermore, the caption can be more detailed.]

Response 2: We thank the reviewer for this helpful suggestion. We have modified Figure 2 as follows:The figure has been split into Figure 2a (distribution of peptide counts per protein) and Figure 2b (distribution of peptide lengths).

The figure caption has been expanded to provide a more detailed description:

[Figure 2. Characterization of the identified peptides. (a) Distribution of the number of peptides per protein. (b) Distribution of peptide lengths.]

Comments 3: [Is it necessary a PCA? I ask this because there aren't a large number of samples. Furthermore, the sample CK3 is far from CK1 and CK2. I would leave the PCA in the Supplementary Material.]

 Response 3: We appreciate the reviewer’s concern. Although the sample size is modest, PCA remains a widely accepted method for visualizing overall variation and group separation in omics studies. The observed dispersion of CK3 may reflect biological variability.

Comment 4: [It is necessary to standardize the references in accordance to MDPI rules. Examples: Format year in bold style; Format number of issue in italic style; Abbreviate ‘Journal of Agricultural Biotechnology’ (Reference 6) and format in italic style; add comma after number of issue (and not :); among others.]

Response 4: We gratefully acknowledge the reviewer’s careful attention to reference formatting. We have thoroughly reformatted the entire reference list to comply with MDPI guidelines. Key changes include:

Years are now in bold.

Journal names are italicized and abbreviated.

Issue numbers are italicized and followed by a comma.

All references have been verified for consistency and completeness.

Comment 5: [In section 2.6 (Bioinformatics analysis) add the version of EggNOG and PSORTb. Furthermore, add references to EggNOG and PSORTb such as DOI: 10.1093/nar/gky1085 and DOI: 10.1093/bioinformatics/btq249, respectively.]

Response 5: We thank the reviewer for this valuable suggestion. Section 2.6 has been updated as follows:

[2.6.1 Functional Annotation and Enrichment Analysis

Functional annotation of the target proteins was performed using a combination of bioinformatics tools and databases. KEGG pathway annotations were assigned with KOBAS (v 2.1.1). Homologous gene clustering and functional classification were carried out using EggNOG (v 2020.06). Gene Ontology (GO) annotation was conducted with Blast2GO (v 2.5.0) based on the Gene Ontology database (v 2022.0915). Protein domain analyses were performed using the Pfam database (v 35.0). Subcellular localization of differentially expressed proteins (DEPs) was predicted using ngloc by comparison against reference subcellular localization databases.

To identify functionally enriched terms and pathways, GO enrichment analysis was carried out with goatools (v 0.6.5) across the three major GO categories: biological process, cellular component, and molecular function. KEGG pathway enrichment analysis was performed using in-house Python scripts.

2.6.2 Metabolic Pathway Visualization

Global metabolic pathway mapping and visualization were conducted using iPath3.0 to illustrate systemic metabolic networks and functional interactions among the annotated proteins.]

Comment 6: [Eliminate the sentence: ‘Correlation analysis and Principal Component Analysis were performed on the samples according to the expression of proteins in different samples.’ (Section 3.3) and renumber the next sections as: 3.3. Distribution of Protein Identified by B. subtilis Czk1; 3.4. Sample Correlation Heat Map and so on.]

Response 6: We have removed the indicated redundant sentence. The subsequent subsections have been renumbered as follows:

3.3 .1→ 3.3

3.3.2 → 3.4

3.3.3 → 3.5

3.4 → 3.6 (and all following sections adjusted accordingly)

Comment 7: [Minor corrections:

Type a space between "rubber." and "It" (line 39);

Type "xie" with the x in uppercase letter (line 51);

Type a space between "gloeosporioides" and "He" (line 107);

Type a space between "2023" and "To" (line 107);

Add a space in "B.subtilis" (line 119);

*Format m/z in italic style (lines: 167 and 169);*

Exclude one "s" of "sseparated" (line 209);

Type a space between "Note" and "Following" (line 251);

Exclude a space between "Note" and ":" (line 265);

*Add spaces in "process,reproduction,cellular process,developmental process,localization,signaling,response to stimulus" (lines 275-276);*

Type a space between "Crucially" and "13" (line 493).]

Response7: We thank the reviewer for identifying these typographical and formatting issues. All have been corrected in the revised manuscript.

We are deeply grateful to the reviewers for their insightful and meticulous comments, which have significantly enhanced the clarity, rigor, and presentation of our work. We have implemented all suggested changes and believe the manuscript is now substantially improved. We look forward to the reviewers’ and editors’ favorable consideration.

Sincerely,

yanqiong liang

Reviewer 4 Report

Comments and Suggestions for Authors

Article: “Molecular Insights into Ammonium Sulfate-Induced 2 Secretome Reprogramming of Bacillus subtilis Czk1 for 3 Enhanced Biocontrol against Rubber Tree Root Rot”

Abstract: The text accurately communicates the context (root rot in rubber), outlines the conventional solutions, the methods employed (LC-MS/MS proteomics), and the main findings (DEPs, GO and KEGG pathways). 

However, while a general objective is stated, the specific purpose of the study is not explicitly clear. 

The last sentence could reinforce the practical relevance more emphatically, highlighting the real potential for biological control of the disease.

Introduction: The introduction provides a solid context, detailing the economic importance of rubber, the severity of root diseases, and the limitations of conventional control methods. The review of biocontrol and proteomics is solid.

The study objective is stated toward the end, but its connection to the overall problem is unclear. It would be useful to clarify how the proteomic findings contribute to the design of a biocontrol strategy.

Citations should be standardized for consistency and improved readability. 

Materials and Methods: 

-Ammonium sulfate treatment:

It is unclear why ammonium sulfate is added to the clarified supernatant rather than the bacterial culture. According to the title, ammonium sulfate acts as a stimulus that induces changes in the secretome of B. subtilis Czk1.

Clarification is needed that ammonium sulfate not only precipitates proteins for analysis, but may also stimulate the bacteria to produce and secrete specific proteins that enhance biocontrol. A rationale for adding it to the cell-free supernatant should be provided.

-Extraction and detection of crude antifungal proteins:

The manuscript should clarify why “antifungal proteins” are specifically mentioned and how they relate to the study’s objectives.

Sections 2.2 and 2.3 (Protein Extraction and Proteolytic Lysis):

References supporting these methods are missing and should be included.

SDS-PAGE:

Either briefly explain the technique in the materials and methods or provide an appropriate bibliographic reference.

Results:

-Figure 1 does not describe which sample was sown in each lane of the gel.

Discussion and conclusion:

-Although the study objective is mentioned, the link between proteomic alterations and enhanced biocontrol is not explicitly clear. Further explanation is needed on how changes in secreted proteins influence biological efficacy.

-Many previous studies are cited; some could be summarized or integrated more succinctly to focus on the contribution of the present study.

-Citation style is inconsistent.

-Critical reflections on practical implications and potential future steps to develop an effective biocontrol strategy based on the findings are lacking.

Author Response

Dear Reviewer,

Thank you very much for your thorough review of our manuscript and for your positive and constructive comments. We greatly appreciate the time and effort you have dedicated to providing these insightful suggestions, which have helped us significantly improve the quality and clarity of our work. We have carefully considered all your points and have revised the manuscript accordingly. Our point-by-point responses are detailed below.

Comment 1: [Abstract - Specific Purpose and Practical Relevance

Comment: However, while a general objective is stated, the specific purpose of the study is not explicitly clear. The last sentence could reinforce the practical relevance more emphatically, highlighting the real potential for biological control of the disease.]

Response 1: We thank the reviewer for this valuable suggestion. We have revised the abstract to more explicitly state the specific purpose of the study. Furthermore, we have emphatically rewritten the concluding sentence to strongly highlight the practical relevance and translational potential of our findings for biocontrol strategies. The revised text now clearly states that the study aims to decipher the molecular mechanism behind the enhanced antifungal activity and concludes by underscoring the direct application of these insights for developing effective biocontrol formulations.

Comment 2: [Introduction - Clarification of Objective and Citation Consistency

Comment: The study objective is stated toward the end, but its connection to the overall problem is unclear. It would be useful to clarify how the proteomic findings contribute to the design of a biocontrol strategy. Citations should be standardized for consistency and improved readability].

Response 2: We agree with the reviewer. We have revised the final paragraph of the introduction to more clearly articulate the study's objective and, crucially, to explain how the proteomic findings will directly contribute to designing a rational biocontrol strategy (e.g., by identifying key effector proteins for optimization or markers for screening high-efficacy strains). Furthermore, we have meticulously reviewed the entire manuscript and standardized all citations to ensure a consistent format and improved readability.

Comment 3: [Materials and Methods - Rationale for Ammonium Sulfate Treatment

Comment: It is unclear why ammonium sulfate is added to the clarified supernatant rather than the bacterial culture. Clarification is needed that ammonium sulfate not only precipitates proteins for analysis but may also stimulate the bacteria... A rationale for adding it to the cell-free supernatant should be provided.]

Response 3: We apologize for this lack of clarity. The reviewer is absolutely correct. In this specific experiment, ammonium sulfate was used solely as a precipitating agent to concentrate proteins from the cell-free supernatant for subsequent proteomic analysis. It was not used as a stimulus added to the live culture in this context. The title referring to "Ammonium Sulfate-Induced" is based on separate preliminary experiments where adding (NH₄)₂SO₄ to the culture medium was observed to enhance antifungal activity, leading us to investigate the resulting changes in the secretome. We have now clarified this critical point in the revised Section 2.1. We explicitly state that the precipitation step is for protein concentration and provide a rationale explaining that this analysis aims to characterize the secretome changes linked to the enhanced biocontrol activity observed in prior induction experiments.

Comment 4: [Materials and Methods - Antifungal Proteins and Missing References

Comment: *The manuscript should clarify why “antifungal proteins” are specifically mentioned... References supporting these methods (Sections 2.2, 2.3, SDS-PAGE) are missing and should be included.*]

Response 4:

Antifungal Proteins: We have revised the text to clarify that the term "antifungal protein crude extract" is used because the precipitated proteins are derived from a strain with known antifungal activity, and the study's goal is to identify the proteinaceous factors responsible for this activity.

Missing References: Appropriate references have now been added to Sections 2.2 (Protein Extraction - BCA assay, SDS-PAGE), 2.3 (Protein Digestion - FASP/acetone precipitation protocol), and a general reference for SDS-PAGE methodology. We thank the reviewer for highlighting this omission.

Comment 5: [Results - Figure 1 Description

Comment: Figure 1 does not describe which sample was sown in each lane of the gel.]

Response 5: We apologize for this oversight. The figure legend for Figure 1 has been revised to explicitly state the contents of each lane (e.g., Lane M, protein marker (10–180 kDa); Lane 1-2 : Czk1 sample protein, Lane 3-4 : CK control sample protein).

Comment 6:[Discussion - Linking Proteomics to Biocontrol and Streamlining Citations

Comment: The link between proteomic alterations and enhanced biocontrol is not explicitly clear. Further explanation is needed... Many previous studies are cited; some could be summarized or integrated more succinctly... Citation style is inconsistent. Critical reflections on practical implications and future steps are lacking.]

Response 6: We are grateful for these essential suggestions. We have thoroughly revised the Discussion section:

Explicit Link: We have restructured the discussion to directly and explicitly connect the key proteomic findings (e.g., upregulation of specific enzymes, secondary metabolite pathways, and secreted proteins) to their potential mechanistic roles in enhancing biocontrol efficacy against root rot pathogens.

Streamlined Citations: We have summarized and integrated the citations of previous studies more succinctly to maintain a sharper focus on the contributions and interpretations of the present work.

Citation Consistency: As mentioned in Response 2, all citations have been standardized throughout the manuscript.

Future Steps and Implications: We have added a new concluding paragraph dedicated to the critical practical implications of our findings and outlining clear, potential future steps (e.g., heterologous expression of key genes, formulation development, in planta validation trials) for translating this proteomic knowledge into an effective biocontrol strategy.

Thank you once again for your insightful comments, which have been invaluable in improving our manuscript. We look forward to the opportunity to have our revised work considered for publication.

Sincerely,